

# Technical Note: Best of both worlds? Combining undisturbed soil monoliths for indoor runoff experiments

David Ramler[1], Peter Strauss[1]

[1]Institute for Land and Water Management Research, Federal Agency for Water Management, Pollnbergstraße 1, Petzenkirchen, A-3252, Austria

*Correspondence to*: David Ramler (david.ramler@baw.at)

**Abstract.** A major decision in soil hydrological research is whether to conduct experiments outdoor or indoors. Both approaches have their advantages and trade-offs. Using undisturbed soil monoliths combines some of the advantages of outdoor and indoor experiments, however, there are often size limitations. While push-methods can be used for small- to medium-sized soil blocks, acquiring larger monoliths necessitates heavy machinery. A promising approach is the combination of smaller blocks to a single large monolith, thereby optimizing cost and labour efficiency as well as representativity and upscaling potential. To this end, we compared the runoff properties of medium-sized (1x 0.5 x 0.35 m) grassland soil monoliths cut in half and re-combined with uncut blocks. We conducted artificial runoff experiments and analyzed the outflow from four flow pathways (surface runoff, subsurface interflow, percolating water, laterally exported water) and surface runoff velocity parameters. Our results suggest that the effects of the re-combination procedure are negligible compared to the variation in the data caused by the inherent soil heterogeneity. Further research is needed for a definite conclusion. Nevertheless, we propose that the benefits of combining soil monoliths outweigh the potential disadvantages.

## 1 Introduction

A cardinal question in soil hydrological research is whether to conduct experiments outdoor or indoor, i.e., *in situ* or *ex situ*. Both are frequently used in artificial rainfall or runoff experiments (e.g., examining erosion or nutrient export) and have each specific strengths and weaknesses. The main advantage of outdoor experiments is that the studied soils have developed naturally and are fully integrated into the surrounding landscape. They are shaped by physico-chemical processes and biological activity and, thus, have developed three-dimensional characteristics that cannot easily be reproduced artificially. Accordingly, the results obtained have an inherent real-life relevance. A downfall is that it can be challenging to find sites with desired conditions, especially ones that are homogeneous over a larger area, so replicate plots can be installed. Frequently, there is only a narrow time frame of constant weather conditions, especially concerning temperature and precipitation (Kuhn et al., 2014).



The main advantage of indoor trials is a higher control over variables and an independency from weather. Furthermore, there is the possibility of testing the same soil under different situations which would be difficult or downright impossible in the field, for example different slopes. Another major benefit is better access to infrastructure, resources, and measuring instruments, saving time and work required (Douglas et al., 1999). Indoor experiments may also be preferred in studies that examine nutrients or pollutants, as these can be collected and discarded without getting into the environment. On the other hand, it is challenging to simulate outdoor conditions with indoor experiments, especially if disturbed soil is used and vegetation is grown artificially (Johnson et al., 1995; Poorter et al., 2016). Owing to limited plot sizes, there is also the question to which extent results can be extrapolated to relevant larger scales (Lewis and Sjöstrom, 2010).

Using undisturbed soil monoliths combines some of the advantages of outdoor and indoor experiments: naturally developed soils combined with high flexibility and control over variables. For upscaling purposes and a more accurate representation of soil processes and their variability, it would be advisable to use as large soil slabs as possible; however, the amount of work needed increases substantially with size. The collection of large monoliths (over 1 m³) necessitates heavy machinery such as hydraulic rams, excavators, and cranes (Belford, 1979; Darch et al., 2015; Schneider et al., 1988). For smaller monoliths push methods can be used, which are usually accomplishable with minimal use of technical gear (Douglas et al., 1999; Palmer et al., 2011). Blocks below 300 kg are easier to handle, store, and discard (Allaire and van Bochove, 2006). Cylindrical soil monoliths are often used for lysimeters while runoff/erosion studies commonly employ rectangular blocks (Allaire and van Bochove, 2006; Douglas et al., 1999). A promising approach for runoff research appears to be the combination of two or more smaller monoliths to a single large block, thereby optimizing cost and labour efficiency as well as representativity and upscaling potential. However, the contact areas between the individual monoliths may affect runoff and transport processes, such as infiltration and sediment movement.

Here, we report on the potential to combine undisturbed soil monoliths to acquire larger soil units for studying runoff and nutrient transport. To this end, we collected six monoliths (1 x 0.5 x 0.35 m) in grassland, representing vegetated filter strips (Prosser et al., 2020). Three monoliths were cut in half and re-combined again, and the others remained uncut. We conducted artificial runoff experiments with tracer applications and flow velocity measurements to examine whether re-combined and uncut blocks behave differently. In principle, the contact zone between two individual blocks could act as a large macropore, promoting preferential flow and, thus, a higher share of percolating water at the expense of surface runoff. However, our main hypothesis was that – done properly – the re-combination procedure has no directional effect on runoff properties. Accordingly, we hypothesized that (1) re-combined monoliths do not differ regarding the (share of) outflow at the different flow pathways, (2) re-combined monoliths do not show a faster onset of percolating water or (3) a faster increase of tracer concentration within the percolating water, and that (4) there is no difference in runoff velocity between treatments. Additionally, we discuss general issues related to indoor runoff experiments.



## 2 Methods

### 2.1 Monolith sampling & preparation

The six monoliths were taken from a permanent grassland near the town of Wieselburg, Lower Austria, Austria (see Table 1 for main soil properties). We used a push method for monolith collection, similarly to the method used in Tiefenbacher et al. (2021). A custom-built steel frame (1 x 0.5 x 0.4 m) with a cutting edge was placed on the soil surface and driven into the ground using body weight and a mallet. To ease penetration and minimize compaction and disturbance, the soil around the frame was gradually removed with spades. Once the desired depth was reached, a bottom plate was inserted with a rack and pinion jack. The frame was towed onto a trailer using wooden ramps and an electrical winch. In the workshop, the monoliths were transferred to plywood boxes. Three of the monoliths were cut in half vertically to obtain two 0.5 x 0.5 x 0.35 m sized blocks. The blocks were interchanged so that the left front of the left block faced the right front of the right block. They were then re-combined by applying a viscous soil-water mixture to the facing fronts and tightening the fit of the plywood box (see Supplement A for details on sampling and cutting). For the soil water mixture, we used soil from the sampling site. Monoliths were watered regularly. However, the soils occasionally dried up to some extent at the surface during hot summer days.

**Table 1: General site characteristics.**

| Soil type | Coordinates | | Annual rainfall | TOC | CaCO$_3$ | pH | Grain size distribution | | |
| --- | --- | --- | --- | --- | --- | --- | --- | --- | --- |
| | Latitude | Longitude | | | | | Clay | Silt | Sand |
| stagni-calcaric cambisol | 48°07'02"N | 15°09'00"E | 700 mm | 1.8 % | < 0.92 % | 6.3 | 38.2 % | 57.3 % | 4.5 % |

### 2.2 Experimental setup

The runoff experiments were carried out in two experimental sets. During the first set, two flow pathways were recorded, the surface runoff (SRF) and subsurface interflow (INT). For the second set, we further sampled and distinguished between percolating water that went through the whole soil body vertically (PER) and laterally exported water (LAT).

The experimental setup consisted of an overflow tank, a steel frame that allowed the collection of surface water, a horizontal plate inserted at the middle of the block to collect subsurface interflow, and for the second set also a bottom steel frame for the collection of percolating and laterally exported water (Fig. 1). The frames were 2 cm smaller than the monoliths on each side, preventing both runoff water from being drained by a gap between the box and the monolith, and laterally exported water to flow into the collector for percolating water. As a precautionary measure, we sealed the potential space between the frame and the soil with a sodium silicate solution ("water glass"; see Supplement B for preliminary experiments on applicability). The slope was adjusted to 3 % during the first set and to 4 % during the second set.





**Fig. 1: Setup of runoff experiments. [A] Overflow tank; [B] metal frame with surface runoff collector; [C] metal plate for subsurface interflow collection; [D] bottom frame with collectors for percolating water (inner outlets) and laterally exported water (outer outlets); [E] rack with slope-adjustable gear.**





Before each experiment, the monoliths were transferred to a water pool until fully saturated and then left to dry for 24 h. Each experimental set comprised three phases. During the first phase, an electric pump distributed runoff over the monolith through the overflow tank. A constant flow of 5 l min$^{-1}$ was adjusted via a valve and water meter. We used deionized water spiked with ortho-phosphate (0.5 mg l$^{-1}$) to mimic agricultural runoff and bromide (~700 mg l$^{-1}$) as a conservative tracer. Outflow was collected and measured using buckets, which were exchanged every minute during the first ten minutes after the onset of an

outflow and afterwards in increasing intervals. Additional samples were taken for chemical analysis after approximately 2, 5, 10, and 30 minutes. Note that the subsurface interflow was very low; as the chemical analyses required a minimum amount of water, taking samples often took substantially longer than the one-minute interval, and not all samples could have been taken. The second phase started after 45 minutes and lasted for approximately 15 minutes, during which surface flow velocity measurements were carried out in three replicates for each monolith. For this, 10 ml of a potassium chloride solution (7.455 g

KCl l$^{-1}$; 12,900 µS) were applied at the upper end of the monolith, and the conductivity at the overflow tank (baseline value) and in the surface runoff collector was monitored (see Supplement C for details). In the last phase, the monoliths were flushed with deionized water for 60 minutes to remove physically retained chemicals.

**2.3 Chemical & statistical analysis**

Water samples were analyzed for bromide and phosphate concentration. Bromide was determined by ion chromatography,

soluble ortho-phosphate was determined photometrically, following national standards.

We conducted non-parametric Kruskal-Wallis tests on the sum of outflow values to check for statistically significant differences between treatments. Bonferroni-adjusted Dunn's post-hoc tests were used to localize significant differences between individual monoliths. We used the quotient of outflow (of the respective flow pathway) to actual inflow, termed standardized outflow, to account for slightly different inflow rates. For the statistics and boxplots, we calculated discharge

values for each flow pathway for every minute from the raw data and only used values between minutes 5 and 45 to eliminate the initial phase where flow rates were not yet stable. For the tracer experiments, we calculated the velocity of the leading edge and the centroid, following Abrantes et al. (2018). Figure generation and statistical testing were carried out using Python 3.9.12 embedded in Spyder 5.1.5 environment. Libraries used were *scipy*, *scikit_posthoc* (statistics), *matplotlib*, *seaborn* (figures), *numpy* and *pandas* (data handling). Statistical significance was set at the $\alpha = 0.05$ level.

**3 Results**

Here, we only report results from the second experimental set. Details on the first set can be found in Supplement D, but are referred to when deemed appropriate.





## 3.1 Water flow

Generally, surface runoff (SRF), percolating water (PER), and laterally exported water (LAT) responded quickly to the runoff
application; the time until an outflow was recorded was commonly around one minute. Both treatments had a similar beginning
of SRF outflow, but re-combined blocks showed an earlier onset of LAT and a later record of PER. Subsurface interflow (INT)
was always the latest to start (Tab. 2).

**Table 2: Results of outflow measurements. Time to runoff gives the time until a runoff was recorded for the respective flow pathway. Share of total water exports gives the percentage that each flow pathway contributes to the total outflow from a monolith. Total water budget is the difference between the applied runoff and the sum of all outflow for each monolith.**

| Position | re-combined | | | | uncut | | | |
|---|---|---|---|---|---|---|---|---|
| | #1 | #3 | #5 | mean | #2 | #4 | #6 | mean |
| Time to runoff [s] | | | | | | | | |
| SRF | 63 | 88 | 55 | 68.7 | 80 | 60 | 67 | 69.0 |
| INT | 420 | 200 | - | 310.0 | 267 | - | 910 | 588.5 |
| PER | 67 | 74 | 60 | 67.0 | 41 | 30 | 25 | 32.0 |
| LAT | 50 | 64 | 40 | 51.3 | 74 | 124 | 69 | 89.0 |
| | | | | | | | | |
| Share of total water export [%] | | | | | | | | |
| SRF | 86.4 | 82.6 | 74.0 | 81.0 | 66.9 | 89.4 | 69.4 | 76.9 |
| INT | < 0.1 | 0.2 | 0.0 | 0.1 | 0.1 | 0.0 | 0.1 | < 0.1 |
| PER | 6.5 | 9.8 | 14.6 | 10.3 | 28.4 | 9.3 | 23.7 | 19.1 |
| LAT | 7.1 | 7.4 | 11.4 | 8.6 | 4.6 | 1.3 | 6.8 | 3.9 |
| | | | | | | | | |
| Total water budget [%] | | | | | | | | |
| | +0.77 | -6.82 | -9.49 | -5.18 | +9.12 | -16.54 | -5.21 | -4.21 |

Irrespective of treatment, SRF always contributed the most to total water outflow at each monolith, followed by PER and LAT.
Generally, INT was very low; the highest share was 0.4 %, while two blocks had no subsurface outflow. At re-combined
monoliths, LAT contributed more to total outflow compared to uncut blocks (Tab. 2).

A similar overall picture was found for standardized outflow, with SRF having the highest outflow, followed by PER, LAT,
and INT (Fig. 2). No significant differences between re-combined and uncut blocks were found for SRF (H = 0.43, P = 0.51)
and PER (H = 2.33, P = 0.13). LAT was slightly below statistical significance (H = 3.86, P = 0.049) and tended to have a
higher outflow (Fig. 2). However, there is an overlap of re-combined and uncut blocks, and post-hoc tests revealed high
heterogeneity in the data. Significant differences between blocks of the same treatment and, *vice versa*, insignificant differences
between blocks of different treatments were found for LAT and all other flow pathways (Supplement E). Statistical testing



was not feasible for INT, due to monoliths with zero outflow. Uncut blocks exhibited substantially higher within-group variances for SRF and LAT.

Four of the six monoliths had a higher water uptake than outflow, but no trend between re-combined and uncut blocks was discernible, again due to substantial inner-group variation (Tab. 2).

### 3.2 Bromide and phosphate

Bromide concentration in the outflow increased with time, approaching 100 % of inflow concentrations. Some blocks also showed bromide concentrations slightly higher than 100 % (Supplement F). Bromide concentrations were high right from the first measurements (i.e., two minutes after the onset of outflow), irrespective of the flow pathway. Lowest initial bromide concentration was 77 % of inflow bromide concentration (block #2, LAT); highest initial concentrations were 98 % (block #5, LAT) and 99 % (block #1, SRF). There was a tendency that phosphate concentrations in the outflow decreased with time when they were initially higher than the inflow phosphate concentration, as well as a tendency to increase when they were lower; in both cases approaching 100 % of the inflow phosphate concentration. Phosphate enrichment in the outflow was substantial, with up to more than twice the inflow concentration for particular samples (Supplement F). No directional difference between re-combined and uncut monoliths was found for neither bromide nor phosphate concentration.

### 3.3 Salt tracer

Although some blocks showed reasonably consistent results in the tracer experiment (e.g. blocks #2-4), there was also substantial inner-block (replicates) and inner-group (treatment) variation (Fig. 3). Consequently, there is much overlap and no directional difference identified between re-combined and uncut monoliths regarding leading-edge or centroid velocity (Supplement G).







**Fig. 2: Outflow rate at the respective flow paths. Green – re-combined blocks, blue – uncut blocks. Different shades denote different blocks. Note that boxplots integrate over minutes 5 to 45. White circles – mean; black line – median; box – 25-75 percentiles; whiskers – 5-95 percentiles; diamonds – outliers.**





Fig. 3: Amplitude passage of salt tracer experiments. Green – re-combined blocks, blue – uncut blocks. Different shades indicate different replicate trials (1-3). Triangles denote timepoint of leading-edge passage; diamonds denote centroids. Black line – quiescent value; dashed line – threshold for leading-edge (see Supplement C for details).



## 4 Discussion

### 4.1 General remarks on indoor runoff experiments

We recommend that soil monoliths are kept outside in a sheltered but sunny location. Blocks need to be watered regularly to avoid drying up – as the blocks are isolated, they are much more prone to detrimental effects than they would be *in situ*. In practice, this can be challenging, for instance over weekends when the workshop or laboratory is vacant. As a general rule, a management plan becomes essential for experimental success if the monoliths are to be kept over a more extended period of time.

Another issue that concerns runoff experiments is the sealing of the soil body. Commonly, the gap between the soil monolith and the box or lysimeter wall is filled using resins, bentonite clay, foams, or other materials. The primary reason for this is to avoid that water is drained via the gap and, thus, does not interact with the soil body (Singh et al., 2018). By filling the gap, the monolith is laterally sealed. In a natural system, a given volume of soil would exchange water with the surrounding soil through the matrix and macropore flow (Beven and Germann, 1982, 2013). In a (more or less) homogeneous environment, lateral water imports and exports from macropore flow would be roughly the same, while the matrix flow would be predominately vertical as the surrounding soil would have the same amount of soil moisture. In a sealed soil monolith, water that macropores would have laterally exported is kept within the soil body while matrix flow is also directed vertically. Therefore, one could argue that sealing is necessary to approximate the water budget of a natural system. In our study, we did not seal the gap between the box and the soil for two reasons: Firstly, a direct drainage of water via the gap is already impeded by the frame which is smaller than the monolith and forces the runoff to flow over – and into – the soil. Secondly, we aimed to mimic a vegetated filter strip (VFS). Despite still prevailing assumptions of runoff occurring as uniform sheet flow, it is, in fact, much more likely that the runoff enters a VFS in concentrated form due to flow convergence (Pankau et al., 2012; Ramler et al., 2022). In this scenario, only the VFS soil under the concentrated flow would receive runoff water, which could then infiltrate into the soil and be laterally exported. In turn, this part of the soil would receive less water from the surrounding soil, which intercepts rain but no runoff water. Accordingly, we propose that this approach provides better conditions for our specific aims. We suggest that future runoff studies ponder whether sealing is appropriate or necessary.

### 4.2 Combining soil blocks

Generally, our main hypothesis that combining monoliths has no directional effect on the runoff properties was supported. Nevertheless, there was a trend of higher LAT outflow at re-combined blocks, accompanied by a faster onset of LAT and a later onset of PER. It appears that the re-combination procedure favours LAT and restricts PER outflow. However, we speculated that a potential difference between treatments would be caused by the contact area between two blocks functioning as an extensive macropore. This would promote preferential flow and quick drainage, leading to a higher amount of percolating and laterally exported water at the expense of surface runoff – which was not the case. One explanation for the higher share of





laterally exported water found for re-combined monoliths would be macropores in the upper half of the contact area and a less permeable lower section which would cause the macropore flow to be diverted sideways. Although this cannot be ruled out, we argue that it is not very probable that this happened. Instead, we suspect that the observed differences are caused by a generally high heterogeneity of the soils, low sample size (n=3), and stochastic effects, e.g., the amount and orientation of

200 macropores such as earthworm channels. Moreover, there was an overlap of standardized outflow values of at least one block from each treatment for all flow pathways. Accordingly, we suggest that the re-combination procedure did not lead to directional differences and, thus, had no adverse effect on runoff properties.

Furthermore, there is no indication to reject the other hypotheses: The re-combined blocks neither showed a faster rise of bromide concentrations nor a slower surface runoff velocity. For most variables, there was a substantial inter-group overlap

and considerable within-group – and in some cases also within-block – variation. Again, this points to a generally high soil heterogeneity, even though all monoliths were taken in close proximity at the same site. Moreover, the water budget of the monoliths and the results from the first experimental set (and partly from the water glass trials) provide similar results; high variation in the data and no directional differences between treatments (Supplement B+D). The direct comparison of both experimental sets further highlights the heterogeneity within the same monolith soil and the influence of repeated experimental

procedures (Darch et al., 2015; Sharpley, 1997). However, this is not a peculiarity of our experiments but rather a common issue in soil research that can only be compensated by increasing the sampling size to average the effects of micro-scale differences in the soil samples (Boix-Fayos et al., 2006; Rüttimann et al., 1995). Nevertheless, this also means that the data noise potentially added by the (re-)combination procedure is probably negligible, suggesting that combining two (or more) blocks is a viable and practicable way to obtain single larger soil monoliths.

Irrespective of treatment, all flow pathways (except the subsurface interflow) had a rapid onset of outflow, commonly around one minute after the start of the experiments. This can only be achieved through preferential macropore flow for the percolating and laterally exported water. This is further backed up by the high amounts of bromide and phosphate already in the first samples (taken after appr. 2 min), which shows that the emerging water originated primarily from the applied runoff and not from the water retained in the soil. As a side note, the enrichment of phosphate found for some blocks also demonstrates that

VFS surface- and subsoils can switch from phosphorous sinks to sources (Andersson et al., 2015; Reid et al., 2018).

After the first set, we were concerned that the high share of percolating water would be due to a flawed setup, causing substantial amounts of water to leak from the system. To this end, we began to apply water glass to the contact areas of the metal frame and the soil and attached an additional metal frame at the bottom to distinguish between actual percolating water and laterally exported water. However, we still recorded a high share of percolating water (between 8 and 27 %) during the

225 second set. Due to the setup, water collected as percolating water must have necessarily travelled through the whole soil body vertically – most probably via macropores – and cannot originate from outside. To shed further light on this issue, we replaced the plywood walls of one box with transparent polycarbonate panes for an additional preliminary trial. We did not conduct a proper runoff experiment as during the first and second set, still, it quickly became apparent that most laterally exported water





outflow was restricted to specific outlets, i.e., through macropore channels that originate within the soil body and lead to the monolith edge (Video 1). Analogously, the same applies to percolating water.

During the first set, the amount of percolating water was high and the onset of outflow was quick. Therefore, the macropores were not caused by drying or re-wetting processes. It is possible, though, that the soil texture was affected by transport and handling. Nevertheless, we argue that the macropores have already been present *in situ*, e.g., amongst others, through a high activity of the soil fauna, as is characteristic for grassland soils (Lamandé et al., 2011; Menta, 2012). Active earthworms were

235 continuously encountered throughout the experiment and were still found nine months after sampling. Wormholes play an essential role in infiltration and can constitute a large share of the macropores within a soil, potentially generating a channel network with high flow rates (Roth and Joschko, 1991; Weiler and Naef, 2003). Furthermore, the collapse or sealing and, *vice versa*, the breakthrough and connection of earthworm channels (and other macropores; Jégou et al., 2002) may explain abrupt changes in outflow seen in some of the blocks (Fig. 2).

After the last experimental set, we left the monoliths without maintenance before they were discarded, which caused them to dry up completely. Thereby, the re-combined monoliths cracked at the contact areas and developed a gap, exemplifying that a complete consolidation (i.e., repair) did not happen. However, this was not expected, as the duration of the experiment – although stretching over several weeks – was too short in relation to the bio-geochemical processes that govern soil development (Pires et al., 2007; Sarmah et al., 1996). Consequently, the contact areas remain a determined breaking point for

drying. A proper merging of individual blocks into a single monolith is probably impossible to achieve in the laboratory. However, we argue that such a high level of 'naturalness' is not necessary for the purpose of runoff experiments; instead, it is sufficient that the combination procedure generates more advantages (e.g., better representativity of processes) than disadvantages (e.g., added data noise).

## 5 Conclusion

Working with undisturbed soil monoliths can be challenging and is always a compromise between available resources and sampling effort (e.g., sampling size, replicates, monolith dimensions). Combining medium-sized monoliths can help to maximize the representativity and upscaling potential of experiments, while minimizing financial and labour efforts. There are, however, some aspects that have to be considered. Proper storage and maintenance are crucial to keeping the monoliths in good condition and are, in turn, dependent on the research aim, the duration of the experiment, climate, and resources (e.g.,

staff, storage space). It is also important that the experimental setup matches the natural hydrological environment of the soil under investigation. In this study, for instance, we refrained from sealing the gap between the soil and box to mimic a grassland soil under concentrated flow (i.e., a vegetated filter strip).

We found general support for our initial hypotheses, as we have no indication that combining two soil monoliths has a directional effect on runoff properties. The observed differences between re-combined and uncut blocks were against

expectations and lacked a clear explanation. We conclude that the inherent heterogeneity of the soils – even if from the same



site – is substantially more considerable and overlays any effect that the combination procedure may have. Accordingly, the advantages outweigh possible adverse effects, and we recommend the use of combined monoliths for indoor runoff studies and related research. Nevertheless, we encourage further research on this subject to better delimit the potential and possible limitations of this procedure, e.g., using X-ray imaging (Bottinelli et al., 2016).

*Code and data availability*. The original contributions presented in the study are included in the article and Supplement; raw data and codes are available upon reasonable request from the corresponding author.

*Supplement*.

*Author contributions*. DR and PS designed the study. DR conducted the experiments, analyzed the data and took the lead in writing and overall organization of the manuscript. All authors contributed to manuscript revision and read and approved the submitted version.

*Competing interests*. The authors declare that they have no conflict of interest.

*Acknowledgements*. The authors thank Günther Schmid and Matthias Karner for assistance during field work, as well as Florian Darmann, Sebastian Rath, Julia Wutzl, Thomas Brunner, and Matthias Konzett for their help with the experiments. We are grateful to the agricultural experimental station in Wieselburg (Landwirtschaftliche Bundesversuchswirtschaften GmbH) for the permission to take soil monoliths on their premises.

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
