# Peer review of "Technical Note: Combining undisturbed soil monoliths for hydrological indoor experiments"

_Hydrology and Earth System Sciences, 2022_

## Referee Comment (RC1)

The submitted manuscript describes a novel experimental approach for combining in-field soil columns to create larger soil monoliths for laboratory study. Runoff properties of soil monoliths are examined using artificial runoff experiments to study outflow and runoff characteristics, suggesting that recombining soil monoliths may be a viable, cost-effective solution for conducting and collecting soil for use in laboratory experiments.

The manuscript is within the journal's scope and aligns with the special issue. The authors should consider revising the manuscript to consider the following comments. I suggest a minor revision.

**General comments:**

Consider revising the title. I don't think the title starting with 'Best of both worlds?' makes sense in the context of the article. I would remove this because it's arbitrary and doesn't refer to the second part of the title. Is this about being the best of both worlds in terms of being a balance between field- and laboratory experiments or being about the recombination of monoliths?

The article has a lot on chemical and statistical analysis, and I think this should be reflected in the title and abstract. Although the article discusses combining soil monoliths, there is a lot of material on chemical composition, so I believe the tracer experiments aspect should be stressed more.

You refer to 'push-methods', but this isn't defined. Some clarity in the abstract, introduction or methodology on what this relates to would benefit the reader.

Having read the full article, I wonder if this work is better defined as a research article than a technical note, given the length of the discussion and appeal to lots of different laboratory experiments. Technical notes are typically a few pages in length. See https://www.hydrology-and-earth-system-sciences.net/about/manuscript_types.html.

**Specific comments:**

**Abstract:**

- Please use the multiplication symbol ($\times$) rather than 'x' in the dimensions of soil monoliths
- Revise 'further research is needed for a definite conclusion' – please elaborate on this.
- I'm not sure the wording 'exported' within 'laterally exported water' makes sense. What does this refer to?

**Introduction:**

- Line 19: Considering revising wording 'cardinal' to 'fundamental'.
- Line 19 – 25: Are there any references which support this or discuss the shortcomings/advantages of laboratory and field experiments (*e.g.* Green, 2014, Modelling Geomorphic Systems: Scaled Physical Models or some of the references in the introduction of Green and Pattison, 2022: Christiansen Revisited)
- Line 23: List these three-dimensional characteristics – are these vegetation, hydrological, soil-mechanics, *etc.*
- I feel as though Line 39, 'The collection of large monoliths (over 1m$^3$) necessitates heavy machinery...' should be stressed in the Abstract, as this is a key reason for the study methods.
- Line 38: 'Advisable to use as large soil slabs as possible' – but this is quite generalised in the context of soil monoliths. How large, and when you refer to a soil 'slab', does

this incorporate depth, width, and length? A reference here would be helpful to appreciate the scale or standard to use here.

- Line 42: 'Blocks below 300 kg are easier to handle…' – a reference here is provided, but more context would be helpful on why this figure. Is this because it can be handled using a particular type of machinery. This could relate to the previous comment – maybe 300 kg is a good balance between being small enough that it is practical and easy to work with and large enough that edge effects are minimised. Some extra discussion on this point would be helpful.
- Line 44: 'Combination of two or more…' – are these equally sized blocks? Change to a 'combination of two or more equally sized, smaller monoliths'. State here that the material should also be comparable, as presumably, you wouldn't want to combine different materials.
- Line 49: Multiplication symbol rather than 'x'. Add in 'equally sized monoliths' before the dimensions.
- Line 54: 'Done properly – the recombination procedure has no directional effect on runoff properties'. Discussing the correct procedure or recombination methods would be beneficial to support this statement.
- Line 55: Wording unclear '…do not differ regarding the (share of) outflow at…'. Please clarify the wording here. I think you are referring to separate outflows on combined monoliths, but the wording could be changed to make this clearer to the reader.
- Line 57: Runoff velocity? Please clarify as this is in the supplementary material to the main text…are you expressing runoff as a speed m s$^{-1}$, or is it discharge (volume/time). This needs to be made more apparent here.

**Methods:**

- Line 71: '…the soils occasionally dried up to some extent at the surface' – this is quite vague. I would consider removing this or putting more detail on this point. Line 85: Why were the slopes of 3% and 4% decided upon, and why did these change between runs.
- Line 77: I don't know whether SRF is a good abbreviation for surface runoff. Also, please define what laterally exported water refers to. I don't think the word 'exported' is clear – maybe this should be reworded to lateral flow or lateral spread/transmission. You mention these abbreviations later in the text (Line 120), so I would either remove these abbreviations from the text/tables altogether or not refer to the longer wording in the text again, as these have already been defined in Line 77. I would recommend removing these abbreviations from the text/tables for clarity.
- Line 85: Why were the slopes of 3% and 4% decided upon, and why did these change between runs.
- Line 90: 'Fully saturated and left to dry for 24 hours' – why? To reach field capacity? Please justify does this relate to a standardised methodology (*i.e.* BS).
- Line 92: 'A constant flow of 5 l/m was adjusted via a valve and water meter' – reword this to ...' was applied using a valve and water meter'.
- Line 92: Remove 'we used' and change it to 'deionised water spiked with ortho-phosphate was applied…'
- Line 93: When you refer to 'mimic agricultural runoff', why these concentrations? Are they specific to regulations in Austria for farming practices?
- Line 106: Reword 'we conducted' and subsequent use of 'we' pronoun.
- Line 108: 'We used the quotient of outflow (of the respective flow pathway) to actual inflow' – please reword for clarity.
- Line 114: Are the scripts in the Appendices/supplementary materials?

**Results:**

- Line 121: How much earlier? State this in the text.
- Line 134: High heterogeneity in the data – what might cause this? Some extra discussion on this would be helpful.

**Discussion:**

- Discussion is good. Because this is a technical note, do you have any 'lessons learned' or reflections that you could add if someone tried to replicate these experiments, how would they be able to improve their design?
- Line 167: 'Blocks need to be watered regularly to avoid drying up' – please state why this is important. Is this because drying up will cause the blocks to dissociate from each other and thus affect the experiments, or is there another reason for this?
- Line 191: 'Nevertheless, there was a trend of higher LAT outflow at re-combined blocks – does this refer to the monolith as a whole or just the interface at the contact area between the combined monoliths.

**Conclusion:**

- The conclusion is reasonable, but given the number of tracer experiments that the monoliths were subjected to in the manuscript results/discussion sections, I would summarise these key findings in the conclusions. Specifically, what were the take-home messages of this study and what are the implications of each experiment? For example, recombining monoliths to conduct salt tracer experiments. Bare in mind that this is a technical note when summarising the conclusions.

**Figures/Tables:**

- Figure 1: Annotate arrows to show slope direction (from overflow tank to outflow at the bottom) for clarity. Also, it would be helpful to annotate key hydrological processes (surface runoff, sub-surface interflow, percolation, laterally exported water) as arrows.
- Table 1: Reword caption 'Site and material characteristics. Add size factions for clay, silt and sand (*i.e.* > 2.0 mm) – are these according to BS or an international specification.
- Table 2: Confusing layout…? Maybe plot the water budget…?

---

## Referee Comment (RC3)

**General comments**

The manuscript presents some hydrological experiments conducted in the laboratory on six undisturbed monoliths. Three of them have been half-cut and recombined. The results suggest that the effects of the re-combination procedure are negligible.

Overall, I think the study is valuable, the experiments are well conducted and the manuscript is well written and structured.

My only main concern is about the experimental design. While the overall research question is about constructive large monoliths e.g., over 1 m³ (L39) that should obtain more accurate representation of soil processes, the experiments are conducted to smaller size blocks (1 x 0.5 x 0.35 m). In addition, only two tests are conducted and with same initial and boundary conditions (constant flow of 5 l min-1 after 24h from fully saturated soil). Finally, as different monoliths are compared, soil heterogeneity affect the interpretation of the results. So, how could we address the questions and reach any conclusions? In contrast, based on the research question and the overall general introduction of the study, I would have expected to see, e.g.: tests conducted on larger combined monoliths (e.g., two blocks of 1 x 0.5 x 0.35 m); tests conducted at the same monolith uncut and cut; tests conducted with different initial and boundary conditions. Field tests over the same area could have also been be performed for comparison.

For this reason, I would strongly suggest to increase the number of experiments to try to derive some more general conclusions. In case these could be not anymore feasable, I suggest to rephrase several parts of the manuscript and weakening some statements as listed in the specific comments below.

**Specific comments (L = Line)**

L1. Title not really meaningful, please consider rephrase. The two worlds should be field and lab but no experiments are actually conducted to answere that a combined monolith is the best of both. See general comment.

L7 I guess there are several major decision in soil hydrology depending on the objectives and expertise. The statement that a major decision in soil hydrological research is whether to conduct experiments outdoor or indoors is in my opinion a bold statement. Please consider rephrasing.

L16. The study can be improved by adding some suggestions about which experiments should be conducted for a definite conclusions, .e.g, block size, initial and boundary condition, comparison to field tests etc.

L19. Is this really a cardinal question? For some reasons one could always be fine with lab or field test. See also comment for L7.

L68 The blocks were interchanged so that the left front of the left block faced the right front of the right block. Why?

L71. Discussion can be extended considering the results we should expect for different soil type

L92. Why this flow? Is this representative of hydrological conditions? Alternative scenarios should have been performed. If not possible, discussion should be extended with suggestions about.

L205. The fact that spatial heterogeneity affects more than cut monolith can not be considered as a proof of the validity of the combined procedure.

L213-214. Combining more monolith has been not proofed to be valid in the present study

L230. What is Video 1?

L244. During drying the cut seems to become relevant. So, it could be argue that also runoff could be affected depending on boundary conditions.

261. Same comment as for L213-214

---

## Author Comment (AC1)

**Response to Reviewer 1**

We would like to thank Reviewer 1 for the comments and remarks which will significantly improve the manuscript. Please find our replies to the general and specific comments, as well as suggestions for revisions, below.

**General comments:**

**R1:** Consider revising the title. I don't think the title starting with 'Best of both worlds?' makes sense in the context of the article. I would remove this because it's arbitrary and doesn't refer to the second part of the title. Is this about being the best of both worlds in terms of being a balance between field- and laboratory experiments or being about the recombination of monoliths?

**Answer:** We agree and suggest changing the title to "*Technical Note: Combining undisturbed soil monoliths for hydrological indoor experiments*".

*R1: The article has a lot on chemical and statistical analysis, and I think this should be reflected in the title and abstract. Although the article discusses combining soil monoliths, there is a lot of material on chemical composition, so I believe the tracer experiments aspect should be stressed more.*

**Answer:** The general aim of this study was to explore the possibility of combining small scale soil monoliths into larger scale soil monoliths for conducting soil hydrological indoor experiments; we understand the tracer experiments as one method among others to study the effects of this procedure. Therefore, we suggest to refrain from adding information about the tracer aspects into the title. However, we intend to stress the chemical aspects a bit more in the Abstract.
Suggested additions would read as follows: "*We conducted artificial runoff experiments and analysed the chemical composition and amount of the outflow from four flow pathways (surface runoff, subsurface interflow, percolating water, lateral flow flow). Furthermore, we studied surface runoff velocity parameters using a salt tracer.*".

*R1: You refer to 'push-methods', but this isn't defined. Some clarity in the abstract, introduction or methodology on what this relates to would benefit the reader.*

**Answer**: Push-methods are opposed to other methods for monolith sampling, such as drilling. Push-methods can be used for large or small monoliths, the difference being the dimensions of the sampling frame and the necessary force needed to push the frame into the ground, as well as the extent of the necessary soil excavation around the sampling frame. For large monoliths a front-end loader may be needed while for smaller monoliths a mallet and spade may be sufficient.
We will add a clarifying sentence to the Introduction that would read as follows: "*For smaller monoliths it is possible to use methods that require only minimal use of technical gear. For instance, push methods can be applied in which a sampling frame is driven into the ground with a mallet or a frame may be built around a pre-cut soil volume*".

*R1: Having read the full article, I wonder if this work is better defined as a research article than a technical note, given the length of the discussion and appeal to lots of different laboratory experiments. Technical notes are typically a few pages in length. See https://www.hydrology-and-earth-system-sciences.net/about/manuscript_types.html.*

**Answer:** We are aware that our manuscript is rather long for a Technical Note. Of course, we tried to be concise, but as we have used a quite extensive approach to examine if what we propose is a valid

method, our manuscript ended up being longer than the average 'Technical Note'. From the HESS manuscript types website:

"**Research articles** report on original research which clearly advances our understanding of hydrological processes and systems, and/or their role in water resources management and Earth system functioning …"

"**Technical notes** report new developments, significant advances, and novel aspects of experimental and theoretical methods and techniques …"

As we see it, we describe a novel experimental method/technique, which could be used for original research. Consequently, we would like to keep this paper as a Technical Note. However, if the Editor or Reviewers have strong objections against this, we are also happy to publish this as a Research Article.

**Specific comments**

**Abstract**

*R1: Please use the multiplication symbol (×) rather than 'x' in the dimensions of soil monoliths*

**Answer:** will be changed.

*R1: Revise 'further research is needed for a definite conclusion' – please elaborate on this.*

**Answer:** We suggest to delete this sentence (also partly due to the word limit for the Abstract). It was supposed to echo the statement we had made at the end of the conclusions – "*Nevertheless, we encourage further research on this subject to better delimit the potential and possible limitations of this procedure, e.g., using X-ray imaging*" – shortened and paraphrased for the Abstract. At a second glance, we think this sentence is probably misleading and actually not really necessary here. We think that the wording (*our results suggest…*, *we propose…*) is defensive enough to clarify that there is still room for improvement.
Nevertheless, we will add a paragraph to the Discussion elaborating on this issue which would read as follows: "*We used an extensive approach, analysing runoff characteristics, chemical loadings, and flow velocity and are, thus, confident in our conclusions. Nevertheless, there is room for further research, for instance applying different boundary and initial conditions, such as other soil types and flow rates, or a higher sample size and number of blocks to be combined. Very interesting would also be a comparison of large monoliths taken with heavy machinery, compared to blocks of similar extent that are made up of combined smaller monoliths.*".

*R1: I'm not sure the wording 'exported' within 'laterally exported water' makes sense. What does this refer to?*

**Answer:** We were referring to water that infiltrated into the soil body but left the soil again (laterally). We intend to change the naming of this flow path to '*lateral flow*' throughout the text.

**Introduction**

*R1:* *Line 19: Considering revising wording 'cardinal' to 'fundamental'.*

**Answer:** We suggest changing the wording to "*important*", also following the suggestions of another reviewer.

*R1:* *Line 19 – 25: Are there any references which support this or discuss the shortcomings/advantages of laboratory and field experiments (e.g. Green, 2014, Modelling Geomorphic Systems: Scaled Physical Models or some of the references in the introduction of Green and Pattison, 2022: Christiansen Revisited)*

**Answer:** Admittedly, we found it more difficult than one would expect to find suitable citations for these seemingly obvious statements. Nevertheless, we agree that backing-up these claims with references would be beneficial and thank the referee for providing two citations. We intend to add at least one of them, together with another reference, more closely related to experimental soil studies (e.g., Katagi, 2013).

*R1:* *Line 23: List these three-dimensional characteristics – are these vegetation, hydrological, soil-mechanics,* etc.

**Answer:** We will list some examples. The proposed change in wording would read as follows: „*They are shaped by physico-chemical processes and biological activity and, thus, have developed three-dimensional characteristics (e.g., vertical gradients, macropore network, root system, etc.) that cannot easily be reproduced artificially.*".

*R1:* *I feel as though Line 39, 'The collection of large monoliths (over 1m3) necessitates heavy machinery...' should be stressed in the Abstract, as this is a key reason for the study methods.*

**Answer:** We intend to rephrase parts of the Abstract to stress this more clearly. The rephrased part of the Abstract would read as follows: "*Acquiring large monoliths necessitates the use of heavy machinery, which is time-, cost- and labour-intensive. Small- to medium-sized soil blocks, however, can be obtained using less demanding methods.*".

*R1:* *Line 38: 'Advisable to use as large soil slabs as possible' – but this is quite generalised in the context of soil monoliths. How large, and when you refer to a soil 'slab', does this incorporate depth, width, and length? A reference here would be helpful to appreciate the scale or standard to use here.*

**Answer**: The range of undisturbed monolith dimensions to be implemented into indoor studies is restricted to a few m width and depth, probably around 10 m² in area and depths < 2.5 m, even if very heavy gear is used. We do not think that it is feasible to provide a number or scale here above which a monolith is supposed to produce results that are representative.
However, we would change the wording here, so that it is clear that all three dimensions are meant and to avoid confusions about the term '*slab*' (which was used as a synonym for monolith/block). The proposed wording would read as follows: "*For upscaling purposes and a more accurate representation of soil processes and their variability, it would be desirable to use soil volumes that are as large as possible; however, the amount of work needed increases substantially with size.*".

*R1:* *Line 42: 'Blocks below 300 kg are easier to handle...' – a reference here is provided, but more context would be helpful on why this figure. Is this because it can be handled using a particular type of machinery. This could relate to the previous comment – maybe 300 kg is a good balance between being small enough*

*that it is practical and easy to work with and large enough that edge effects are minimised. Some extra discussion on this point would be helpful.*

**Answer**: The 300 kg were taken from the reference. We realise that this number appears to be quite arbitrary without context. We intend to rephrase the sentence to a more generalised statement, e.g.; "*Furthermore, the less heavy the monoliths are, the easier they are to handle, store, and discard*.". We do not think that the manuscript would be improved by further discussion. It is obvious that smaller and lighter monolith are easier to handle and store, but how easy the handling is also depends on the resources of the laboratory (access to cranes, lift trucks, manpower, etc.).

*R1: Line 44: 'Combination of two or more...' – are these equally sized blocks? Change to a 'combination of two or more equally sized, smaller monoliths'. State here that the material should also be comparable, as presumably, you wouldn't want to combine different materials.*

**Answer:** We were thinking that the monoliths would all be taken from the same site and with the same sampling gear and method, thus being equally sized. We will rephrase the sentence to include this, which would read as follows: "*A promising approach for runoff research appears to be taking two or more equally sized smaller monoliths at a site and combining them to a single large block, thereby optimizing cost and labour efficiency as well as representativity and upscaling potential*.".

*R1: Line 49: Multiplication symbol rather than 'x'. Add in 'equally sized monoliths' before the dimensions.*

**Answer:** will be changed accordingly.

*R1: Line 54: 'Done properly – the recombination procedure has no directional effect on runoff properties'. Discussing the correct procedure or recombination methods would be beneficial to support this statement.*

**Answer:** We agree and will add some more details on a '*proper re-combination*' in the M&M section. The new wording would read as follows: "*For a proper re-combination, it is essential that the blocks are of equal width and depth. Furthermore, the monoliths, and the soil used for the soil-water mixture, need to be taken from the same site in close proximity to one another*.".

*R1: Line 55: Wording unclear '...do not differ regarding the (share of) outflow at...'. Please clarify the wording here. I think you are referring to separate outflows on combined monoliths, but the wording could be changed to make this clearer to the reader.*

**Answer:** What we wanted to express is that re-combined monoliths do not differ regarding the amount of outflow of the different flow pathways, as well as their proportional share. We will change the wording here for clarification into: "*Accordingly, we hypothesized that (1) re-combined monoliths do not differ regarding the amount of outflow at the different flow pathways and their proportional share, [...]*".

*R1: Line 57: Runoff velocity? Please clarify as this is in the supplementary material to the main text...are you expressing runoff as a speed m s-1, or is it discharge (volume/time). This needs to be made more apparent here.*

**Answer**: The unit of a velocity is distance per time (usually $m\ s^{-1}$), the unit of a discharge is volume per time. As we speak of '*runoff velocity*' we do not think that there is an ambiguity. However, we will add the unit for clarity.

**Methods**

*R1: Line 71: '…the soils occasionally dried up to some extent at the surface' – this is quite vague. I would consider removing this or putting more detail on this point.*

**Answer:** We suggest to remove this sentence.

*R1: Line 77: I don't know whether SRF is a good abbreviation for surface runoff. Also, please define what laterally exported water refers to. I don't think the word 'exported' is clear – maybe this should be reworded to lateral flow or lateral spread/transmission. You mention these abbreviations later in the text (Line 120), so I would either remove these abbreviations from the text/tables altogether or not refer to the longer wording in the text again, as these have already been defined in Line 77. I would recommend removing these abbreviations from the text/tables for clarity.*

**Answer:** We agree. We intend to remove all abbreviations for the flow pathways and will instead refer to the full wording throughout the text and tables.
We would also add a better description for laterally exported water, which would read as follows: "*The runoff experiments were carried out in two experimental sets. During the first set, two flow pathways were recorded at the lower end of the monolith, the surface runoff and subsurface interflow. For the second set, we further sampled and distinguished between percolating water that went through the whole soil body vertically and water that infiltrated into the soil body, but left the monolith again at the side due to lateral flow pathways.*".
Furthermore, we will change the wording from "laterally exported water" to "*lateral flow*" throughout the text as suggested.

*R1: Line 85: Why were the slopes of 3% and 4% decided upon, and why did these change between runs.*

**Answer**: Based on our data from established VFS in Austria, 3-4 % are commonly found slopes of VFS. We will add some text to clarify this, which would read as follows: "*The inclination was based on typical slopes of VFS in Austria and were adjusted to 3 % during the first set and 4 % during the second set.*". The first monolith of the second set was adjusted to 4 % because we erroneously thought we also used 4 % during the first set. The 4 % were than kept for the rest of the second set to have equal conditions within a set. As the switch from 3 to 4 % is rather negligible, we would refrain from further explanations in the text.

*R1: Line 90: 'Fully saturated and left to dry for 24 hours' – why? To reach field capacity? Please justify does this relate to a standardised methodology (i.e. BS).*

**Answer**: The monoliths were saturated (and left to drain) in order to reach comparable soil moisture levels for all monoliths. The aim of the drainage procedure was to obtain conditions of field capacity for the soil water content. Previous work has shown, that for this type of soil monoliths (soil depths of about 30 cm), field capacity is reached after 1 day of free drainage (Tiefenbacher et al., 2021).
We will add a clarification, which would read as follows: "*Before each experiment, the monoliths were transferred to a water pool until fully saturated and then left to drain for 24 h, to obtain comparable conditions of field capacity for the soil water content (see Tiefenbacher et al., 2021).*".

*R1: Line 92: 'A constant flow of 5 l/m was adjusted via a valve and water meter' – reword this to …' was applied using a valve and water meter'.*
*Line 92: Remove 'we used' and change it to 'deionised water spiked with ortho-phosphate was applied…'*

**Answer:** We will change both sentences as suggested.

*R1: Line 93: When you refer to 'mimic agricultural runoff', why these concentrations? Are they specific to regulations in Austria for farming practices?*

**Answer**: This concentration was based on a previous (unpublished) study, which analysed the runoff from fields using artificial rainfall. Data for typical (Lower) Austrian agricultural soils ranged from 0.2 – 0.9 mg DP $l^{-1}$. The used value of 0.5 mg $l^{-1}$ is more or less a mean value. We will add the wording "*typical local*" to clarify this a bit better.

*R1: Line 106: Reword 'we conducted' and subsequent use of 'we' pronoun.*

**Answer:** We will change the wording accordingly.

*R1: Line 108: 'We used the quotient of outflow (of the respective flow pathway) to actual inflow' – please reword for clarity.*

**Answer:** Sentence will be rephrased for clarity and would read as follows: "*To account for slightly different inflow rates between monoliths, a standardized outflow value was used, which was calculated by dividing the measured outflow rate at a flow pathway by the actual inflow rate.*".

*R1: Line 114: Are the scripts in the Appendices/supplementary materials?*

**Answer**: We did not introduce any new formulas or functions. We only used already existing libraries and the functions therein. Thus, we do not see much benefit for the reader of including the scripts to the Appendix. Nevertheless, as stated in the Section "*Code and data availability: [..] raw data and codes are available upon reasonable request from the corresponding author.*"

**Results**

*R1: Line 121: How much earlier? State this in the text.*

**Answer:** We will add mean numbers to the text. The rephrased text would read as follows: "*Both treatments had a similar beginning of surface runoff outflow at around 69 s, but re-combined blocks had an on average 38 s earlier onset of lateral flow and 35 s later onset of percolating water.*".

*R1: Line 134: High heterogeneity in the data – what might cause this? Some extra discussion on this would be helpful.*

**Answer:** We will add a more detailed discussion on the sources of soil variability to the Discussion section 4.2. The addition would read as follows: "*Sources of this natural variability of the soil are manifold, including vegetation patterns, edaphon activity, (micro-)relief, soil aggregation, and their often complex interactions with runoff (Boix-Fayos et al., 2006; Bryan and Luk, 1981). Furthermore, there may be anthropogenic impacts before, during, and after the monolith collection (Luk and Morgan, 1981; Rüttimann et al., 1995).*".

**Discussion**

*R1: Discussion is good. Because this is a technical note, do you have any 'lessons learned' or reflections that you could add if someone tried to replicate these experiments, how would they be able to improve their design?*

**Answer:** Some considerations can already be found in the Discussion section 4.1, to which we would add additional paragraphs (e.g., regarding what is necessary for a successful combination; different soil types, etc.).

The intended added paragraph would read as follows: "*For a successful combination, it is vital that the single blocks they are of equal width and height. While the width is usually fixed by the sampling device, acquiring similar heights may be more difficult. Uneven soil surfaces produce small barriers or ridges when monoliths are combined, which can interfere with runoff patterns. Furthermore, monoliths will have different heights at the front and back end if they are not taken at a right angle to the soil slope. Some adjustments can be made in the laboratory (e.g., different bottom plates to even out height differences). Nevertheless, we strongly advise to avoid such issues in the first place by an a priori assessment of site conditions and careful sampling. This also includes a sampling design that allows the extraction of monoliths in close proximity to another, limiting the effects of larger scale gradients.*". Note that we also intend to delete some sentences and paragraphs from the discussion following suggestions by other reviewers.

*R1: Line 167: 'Blocks need to be watered regularly to avoid drying up' – please state why this is important. Is this because drying up will cause the blocks to dissociate from each other and thus affect the experiments, or is there another reason for this?*

**Answer**: Drying would change the porosity within the soil, which affects runoff properties, but would also impede a proper 'repair' of the contact zone in combined monoliths. In addition, it will negatively affect plant growth of the monoliths.

We would add a sentence to clarify this, which would read as follows: "*Blocks need to be watered regularly for plant vitality and to avoid drying and an emergence of cracks in the soil structure that would affect runoff properties and impede a repair of the combined monoliths.*".

*R1: Line 191: 'Nevertheless, there was a trend of higher LAT outflow at re-combined blocks – does this refer to the monolith as a whole or just the interface at the contact area between the combined monoliths.*

**Answer**: This refers to the whole (re-combined) block. We can only speculate what happens at the interface at the contact area (what we do in the following paragraph).

**Conclusion**

*R1: The conclusion is reasonable, but given the number of tracer experiments that the monoliths were subjected to in the manuscript results/discussion sections, I would summarise these key findings in the conclusions. Specifically, what were the take-home messages of this study and what are the implications of each experiment? For example, recombining monoliths to conduct salt tracer experiments. Bare in mind that this is a technical note when summarising the conclusions.*

**Answer**: We understand that the importance of our study is the finding that the method we used is a valid procedure. In this respect, the results of the different experiments are just a way to test the procedure and are not of primary interest as such. We have structured the take-home messages: in the first paragraph, we state what is important for the sampling and combination procedure; in the second paragraph, we describe why we think the results support the application of this method. Thus, we still think that the take-home messages are outlined sufficiently and are fit for a Technical Note. If we are missing something, we kindly ask R1 for further clarifications.

**Figures/Tables**

*R1: Figure 1: Annotate arrows to show slope direction (from overflow tank to outflow at the bottom) for clarity. Also, it would be helpful to annotate key hydrological processes (surface runoff, sub-surface interflow, percolation, laterally exported water) as arrows.*

**Answer:** We will change Fig. 1 accordingly, e.g.:

[Figure]

*Figure 1: Fig. 1: Setup of runoff experiments with flow pathways (arrows). [A] Overflow tank; [B] metal frame with surface runoff (RUN) collector; [C] metal plate for subsurface interflow (INT) collection; [D] bottom frame with collectors for percolating water (PER; inner outlets) and lateral flow (LAT; outer outlets); [E] rack with slope-adjustable gear.*

*R1: Table 1: Reword caption 'Site and material characteristics. Add size factions for clay, silt and sand (i.e. > 2.0 mm) – are these according to BS or an international specification.*

**Answer:** We will reword the caption and add size factions. The size factions are based on national standards (ÖNORM). The revised table header would read as follows: "*Table 1: Site and material characteristics. TOC – total organic carbon; CaCO$_3$ – calcium carbonite. Coarse material > 2 mm; Sand 2–0.063 mm; Silt 0.063–0.002 mm; Clay < 0.002 mm.*".

*R1:* *Table 2: Confusing layout…? Maybe plot the water budget…?*

**Answer**: Plotting the water budget would result in an additional figure. We think that the advantage of providing this information in one table row is about saving space. Thus, we suggest keeping the table as it is.

---

## Author Comment (AC2)

**Response to Reviewer 2**

We would like to thank Reviewer 2 for the comments and remarks which will significantly improve the manuscript. Please find our replies to the general and particular comments, as well as suggestions for revisions, below.

*R2: Soil characteristics (Table 1): TOC is undefined. Key soil characteristics such as porosity, dry density, gravel content, organic matter should be given if they are available. Could this method be applied to other soil types?*

**Answer:** We will define TOC and $CaCO_3$ and revise the table header, also following suggestions from another reviewer. The revised table header would read as follows: "*Table 1: Site and material characteristics. TOC – total organic carbon; $CaCO_3$ – calcium carbonite. Coarse material > 2 mm; Sand 2–0.063 mm; Silt 0.063–0.002 mm; Clay < 0.002 mm.*".

We will add information on coarse material (> 2 mm). Unfortunately, we cannot provide data on the other requested soil characteristics.

In general, this method should be applicable to a rather broad spectrum of soil textures, which is the main criteria for application. However, we do not have information yet to add textural limits, but we suppose that soils with a very sandy texture will be difficult to handle, because of their weak structural integrity.

We will add a discussion to section 4.1., which would read as follows: "*In principal, we expect that the combination method is applicable to a large variety of soil types, but cannot yet provide textural limits. However, combining two soil blocks requires removing the bordering at one side. Thus, soils that are structurally weak and could collapse without a frame are not suitable for this method.*".

*R2: Statistical analysis: differences between experiments are analyzed using statistical metrics. The Methods section does not clearly explain how this is done. For example, H and P (Line 132) are undefined. L. 114: Significance of what? What would have been the results at the 0.01 level?*

**Answer:** *H* and *P* are the outcomes of the Kruskal-Wallis test: *H* is the test statistic, *P* is the probability measure. These two abbreviations are commonly used and we suggest not to give definitions for these test statistic or p-value.

Statistical significance was set to 5 %, as is usually the case in virtually all studies across various scientific fields. A significance level, or $\alpha$, of 0.05 refers to a risk of 5 % that we conclude we have found significant differences between two (or more) groups when in fact there are none. This value is common practice, although, of course, arbitrary and sometimes criticized. We suggest not to deviate from it. Nevertheless, in the manuscript we do not solely rely on statistical significance alone, but also discuss the meaningfulness of found differences (e.g., if there is a directional effect or trend, how substantial the variation is, etc.). Our conclusions would have been no other if $\alpha$ would have been set at 0.001 (or any other value, for that matter).

We hope that we did not misunderstand the intention of R2 in this comment; if so, we would kindly ask R2 to clarify the issue.

*R2: Experimental design is unclear. For example, on L. 116, a "second experimental set" is mentioned. What is this? The first line of Table 2 is not complete for understanding the experimental plan. A new Table listing all experiments would be useful.*

**Answer:** The "*second experimental set*" is the one that is described in the manuscript; data on the first experimental set is provided in the Supplement. Both are described at the beginning of section 2.2. Maybe the wording ("*set*") is misleading? What we meant is synonymous to "*trial*" or "*round*".

During the first set, we conducted experiments on all 6 monoliths and recorded the outflow at 2 flow pathways. After some improvement of the overall set-up, the second set started, during which we conducted another round of experiments on all 6 monoliths, but this time recording the outflow at 4 flow pathways. As the latter is more elaborate, we decided to only report on the results of the second set in the main text. However, as stated in the discussion, the first set provides similar results. Thus, as long as not explicitly stated, all results in the main text refer to the second experimental set.

To clarify this, we will rephrase the paragraph, which would read as follows: "*The runoff experiments were carried out in two experimental sets, each comprising a full round of experiments (runoff and tracer measurements) on all six monoliths. During the first set, two flow pathways were recorded at the lower end of the monolith, the surface runoff and subsurface interflow. For the second set, we further sampled and distinguished between percolating water that went through the whole soil body vertically and water that infiltrated into the soil body, but left the monolith again at the side due to lateral flow pathways.*".

Maybe we are missing the point; in this case we would kindly ask R2 for a clarification.

*R2: Discussion: Section 4.2 is too long, especially for a technical note. Could be more concise.*

**Answer**: We agree that the discussion is long, especially for a Technical Note. This is mainly due to the fact the we used a rather extensive approach (outflow measurements, chemical loading of runoff, flow velocity). We will shorten and rephrase the discussion where possible to be more concise. Parts of the Discussion will also be changed/rephrased following suggestions from other reviewers.

*R2: The title of the paper could be improved.*

**Answer**: We agree. We suggest to change the title to "*Technical Note: Combining undisturbed soil monoliths for hydrological indoor experiments*".

---

## Author Comment (AC3)

**Response to Reviewer 3**

We would like to thank Reviewer 3 for the comments and remarks which will significantly improve the manuscript. Please find our replies to the general and specific comments, as well as suggestions for revisions, below.

**General comment / main concern:**

*R3: My only main concern is about the experimental design. While the overall research question is about constructive large monoliths e.g., over 1 m³ (L39) that should obtain more accurate representation of soil processes, the experiments are conducted to smaller size blocks (1 x 0.5 x 0.35 m). In addition, only two tests are conducted and with same initial and boundary conditions (constant flow of 5 l min-1 after 24h from fully saturated soil). Finally, as different monoliths are compared, soil heterogeneity affect the interpretation of the results. So, how could we address the questions and reach any conclusions? In contrast, based on the research question and the overall general introduction of the study, I would have expected to see, e.g.: tests conducted on larger combined monoliths (e.g., two blocks of 1 x 0.5 x 0.35 m); tests conducted at the same monolith uncut and cut; tests conducted with different initial and boundary conditions. Field tests over the same area could have also been be performed for comparison. For this reason, I would strongly suggest to increase the number of experiments to try to derive some more general conclusions. In case these could be not anymore feasable, I suggest to rephrase several parts of the manuscript and weakening some statements as listed in the specific comments below.*

**Answer:** In general, we agree with what the reviewer states in this comment. Conducting experiments on larger monoliths, using different initial and boundary conditions, or the inclusion of field tests would have all been beneficial to support our hypothesis. However, we see no solution to the problem of soil heterogeneity other than repeating experiments with a high number of replicates. In this way, we may at least find out if soil heterogeneity is larger than procedural heterogeneity (which we did). Please be aware, that using the same soil monoliths uncut and cut will also introduce heterogeneity of unknown size due to the handling necessary during the resampling procedure.

Indeed, as R3 already mentions, conducting these additional experiments was/is not feasible. In fact, our experimental set-up presented in this manuscript (6 monoliths, 2 experimental sets, analysis of runoff characteristics, chemical properties, and flow velocity, 4 flow pathways) was already quite time-consuming and labour-intensive. While we agree that the suggested additional experiments are reasonable additions and something that may be done in future studies, we think that our results are sufficiently informative to draw the conclusion that combining monoliths is a sound procedure. However, we would discuss this issue in more detail in the manuscript and would rephrase several parts of the manuscript as suggested by R3 and the other reviewers (see Responses to Specific Comments).

**Specific comments**

*R3: Title not really meaningful, please consider rephrase. The two worlds should be field and lab but no experiments are actually conducted to answere that a combined monolith is the best of both. See general comment.*

**Answer:** We would rephrase the title to "*Technical Note: Combining undisturbed soil monoliths for hydrological indoor experiments*".

*R3:* *L7 I guess there are several major decision in soil hydrology depending on the objectives and expertise. The statement that a major decision in soil hydrological research is whether to conduct experiments outdoor or indoors is in my opinion a bold statement. Please consider rephrasing.*

**Answer**: We agree that the decision whether to conduct the experiment under field or laboratory conditions is one of several options in soil hydrology. We will rephrase to: "*An important decision […]*", to better reflect this. See also response to Comment L19.

*R3:* *L16. The study can be improved by adding some suggestions about which experiments should be conducted for a definite conclusions, e.g, block size, initial and boundary condition, comparison to field tests etc.*

**Answer**: We suggest to delete this sentence in Line 16 from the Abstract (following other reviewers' suggestions). We think that the wording (*our results suggest…, we propose…*) is defensive enough to clarify that there is still room for improvement.
Nevertheless, we intent to discuss this issue in more detail in section 4.2., by adding a paragraph that would read as follows: "*We used an extensive approach, analysing runoff characteristics, chemical loadings, and flow velocity and are, thus, confident in our conclusions. Nevertheless, there is room for further research, for instance applying different boundary and initial conditions, such as other soil types and flow rates, or a higher sample size and number of blocks to be combined. Very interesting would also be a comparison of large monoliths taken with heavy machinery, compared to blocks of similar extent that are made up of combined smaller monoliths.*".

*R3:* *L19. Is this really a cardinal question? For some reasons one could always be fine with lab or field test.*

**Answer**: We agree, that one could always be fine with lab or field tests. However, each comes with specific advantages and drawbacks and they are, thus, not interchangeable and dependent on the research question and resources. The rather strong (original) wording of this sentence has also been driven by the fact that in soil hydrology (as opposed to other hydrological fields) the question of conducting indoor or outdoor studies is quite often an issue.
To account for this comment, we suggest to change the wording from "*cardinal*" to "*important*".

*R3:* *L68 The blocks were interchanged so that the left front of the left block faced the right front of the right block. Why?*

**Answer**: Blocks were interchanged to better mimic the combination of two separately taken monoliths. If we would have combined the separated monoliths again directly at the cut without interchanging, we were concerned that this would lead to a "perfect fit" (e.g., concerning macropore channels) that would not be possible to obtain when separately taken monoliths are used.

*R3:* *L71. Discussion can be extended considering the results we should expect for different soil type*

**Answer**: We will add a discussion on different soil types at section 4.1., which would read as follows: "*In principal, we expect that the combination method is applicable to a large variety of soil types, but cannot yet provide textural limits. However, combining two soil blocks requires removing the bordering at one side. Thus, soils that are structurally weak and could collapse without a frame are not suitable for this method.*".

*R3: L92. Why this flow? Is this representative of hydrological conditions? Alternative scenarios should have been performed. If not possible, discussion should be extended with suggestions about.*

**Answer**: This flow rate was chosen based on previous experiments We know from previous (unpublished) studies on similar soils that a large (though not extreme) rainfall event of 60 mm would produce a runoff within the range of 0.1 l min$^{-1}$ m$^{-2}$. The chosen 5 l min$^{-1}$ would therefore correspond to a source area of 500 m² that contributes to a concentrated flow at the field edge – which is probably at the lower end of realistic source areas. Other studies using artificial runoff (e.g., Guertault et al. 2021, Saleh et al. 2017) used much higher flow rates (12-99, and 99 l min$^{-1}$, respectively). This would not be possible to handle with our set-up. Nevertheless, we also chose a lower flow rate to encourage infiltration, as we speculated that the cut could act as a large macropore. Higher flow rates and, thus, higher flow velocities would promote surface runoff at the expense of infiltration.
Suggestions on experiments with different flow rates would be included in the added paragraph referred to in the Response to Comment L16 (see above).

*R3: L205. The fact that spatial heterogeneity affects more than cut monolith can not be considered as a proof of the validity of the combined procedure.*

**Answer**: We agree that the high heterogeneity encountered is not a proof of the validity of the combination procedure. It is still possible that there is an effect of the combination, which would probably require a (maybe unfeasibly) high number of plots to find out. Based on our data, even if there is an effect it would be small in comparison to the soil heterogeneity. Based on our findings, we only state that "… *there is no indication to reject the hypotheses*", which is true. Thus, we think that our conclusion that the advantages outweigh potential disadvantages of added data noise is supported.

*R3: L213-214. Combining more monolith has been not proofed to be valid in the present study*

**Answer**: We agree, and suggest to rephrase and weaken this statement as follows: "*We conclude that the re-combination procedure did not lead to directional differences and, thus, had no adverse effect on runoff properties, suggesting that combining two (and probably more) blocks is a viable and practicable way to obtain single larger soil monoliths.*".

*R3: L230. What is Video 1?*

**Answer**: Video 1 is part of the *Assets*. The *Assets* consist of a *Supplement* and a *Video supplement* and should both be downloadable at the Preprint Website (Copernicus Office). This short video shows a concentrated outflow at a single outlet, probably an earthworm channel.

*R3: L244. During drying the cut seems to become relevant. So, it could be argue that also runoff could be affected depending on boundary conditions.*

**Answer**: We agree; consequently, it is important that the monoliths are properly maintained, especially regarding regular watering. Experiments should not be carried out on dry-ish soil conditions. We suggest to add clarifications such as: "*We recommend that soil monoliths are kept outside in a sheltered but sunny location. Blocks need to be watered regularly for plant vitality and to avoid drying and an emergence of cracks in the soil structure that would affect runoff properties and impede a repair of the combined monoliths (Bottinelli et al., 2016; Pires et al., 2007)*", to the Discussion and Conclusion.

*R3: 261. Same comment as for L213-214*

**Answer**: We think R3 is referring to the sentence "… *we recommend the use of combined monoliths* …". We admit that the term "*combined monoliths*" may be a bit ambiguous if two or more monoliths are meant. We suggest to leave this unchanged, but we will add some more information in the following sentence to clarify this issue, which would read as follows: "*Nevertheless, we encourage further research on this subject to better delimit the potential and possible limitations of this procedure, for instance using different experimental set-ups (e.g., number of monoliths), boundary conditions (e.g., flow rates, soil types, dimensions), or analysis methods (e.g., X-ray imaging).*".

---

## Author Comment (AC4)

**Response to Reviewer 1**

We would like to thank Reviewer 1 for the comments and remarks which significantly improved the manuscript. Please find our revisions and replies to the general and specific comments below. Line numbers in blue square brackets refer to the line number in the uploaded revised manuscript with tracked changes.

**General comments:**

R1: Consider revising the title. I don't think the title starting with 'Best of both worlds?' makes sense in the context of the article. I would remove this because it's arbitrary and doesn't refer to the second part of the title. Is this about being the best of both worlds in terms of being a balance between field- and laboratory experiments or being about the recombination of monoliths?

**Answer:** We agree and suggest changing the title to "*Technical Note: Combining undisturbed soil monoliths for hydrological indoor experiments*". [Lines 1-2]

*R1: The article has a lot on chemical and statistical analysis, and I think this should be reflected in the title and abstract. Although the article discusses combining soil monoliths, there is a lot of material on chemical composition, so I believe the tracer experiments aspect should be stressed more.*

**Answer:** The general aim of this study was to explore the possibility of combining small scale soil monoliths into larger scale soil monoliths for conducting soil hydrological indoor experiments; we understand the tracer experiments as one method among others to study the effects of this procedure. Therefore, we suggest to refrain from adding information about the tracer aspects into the title. However, we stressed the chemical aspects a bit more in the Abstract: "*We conducted artificial runoff experiments and analysed the chemical composition and amount of the outflow from four flow pathways (surface runoff, subsurface interflow, percolating water, lateral flow flow). Furthermore, we studied surface runoff velocity parameters using a salt tracer.*". [Lines 15-17]

*R1: You refer to 'push-methods', but this isn't defined. Some clarity in the abstract, introduction or methodology on what this relates to would benefit the reader.*

**Answer**: Push-methods are opposed to other methods for monolith sampling, such as drilling. Push-methods can be used for large or small monoliths, the difference being the dimensions of the sampling frame and the necessary force needed to push the frame into the ground, as well as the extent of the necessary soil excavation around the sampling frame. For large monoliths a front-end loader may be needed while for smaller monoliths a mallet and spade may be sufficient.
We added a clarifying sentence to the Introduction that reads as follows: "*For smaller monoliths it is possible to use methods that require only minimal use of technical gear. For instance, push methods can be applied in which a sampling frame is driven into the ground with a mallet, or a frame may be built around a pre-cut soil volume*". [Lines 44-47]

*R1: Having read the full article, I wonder if this work is better defined as a research article than a technical note, given the length of the discussion and appeal to lots of different laboratory experiments. Technical notes are typically a few pages in length. See https://www.hydrology-and-earth-system-sciences.net/about/manuscript_types.html.*

**Answer:** We are aware that our manuscript is rather long for a Technical Note. Of course, we tried to be concise, but as we have used a quite extensive approach to examine if what we propose is a valid method, our manuscript ended up being longer than the average 'Technical Note'. From the HESS manuscript types website:

> "**Research articles** report on original research which clearly advances our understanding of hydrological processes and systems, and/or their role in water resources management and Earth system functioning ..."

> "**Technical notes** report new developments, significant advances, and novel aspects of experimental and theoretical methods and techniques ..."

As we see it, we describe a novel experimental method/technique, which could be used for original research. Consequently, we would like to keep this paper as a Technical Note. However, if the Editor or Reviewers have strong objections against this, we are also happy to publish this as a Research Article.

**Specific comments**

**Abstract**

*R1:* Please use the multiplication symbol (×) rather than 'x' in the dimensions of soil monoliths

**Answer:** Changed. [Line 14]

*R1:* Revise 'further research is needed for a definite conclusion' – please elaborate on this.

**Answer:** We deleted this sentence (also partly due to the word limit for the Abstract). It was supposed to echo the statement we had made at the end of the conclusions – "*Nevertheless, we encourage further research on this subject to better delimit the potential and possible limitations of this procedure, e.g., using X-ray imaging*" – shortened and paraphrased for the Abstract. At a second glance, we think this sentence is probably misleading and actually not really necessary here. We think that the wording (*our results suggest*…, *we propose*…) is defensive enough to clarify that there is still room for improvement. [Lines 17-19]

Nevertheless, we will add a paragraph to the Discussion elaborating on this issue which would read as follows: "*We used a comprehensive approach, analysing runoff characteristics, chemical loadings, and flow velocity and are, thus, confident in our conclusions. Nevertheless, there is room for further research, for instance applying different boundary and initial conditions, such as other soil types and flow rates, or a higher sample size and number of blocks to be combined. A comparison of large monoliths taken with heavy machinery with blocks of similar extent that are made up of combined smaller monoliths would be very interesting.*". [Lines 297-301]

*R1:* I'm not sure the wording 'exported' within 'laterally exported water' makes sense. What does this refer to?

**Answer:** We were referring to water that infiltrated into the soil body but left the soil again (laterally). We changed the naming of this flow path to '*lateral flow*' throughout the text. [Lines 16, 89, ff.]

**Introduction**

*R1:* Line 19: Considering revising wording 'cardinal' to 'fundamental'.

**Answer:** We suggest changing the wording to "*important*", also following the suggestions of another reviewer. [Line 22]

*R1:* Line 19 – 25: Are there any references which support this or discuss the shortcomings/advantages of laboratory and field experiments (e.g. *Green, 2014, Modelling Geomorphic Systems: Scaled Physical Models or some of the references in the introduction of Green and Pattison, 2022: Christiansen Revisited*)

**Answer:** Admittedly, we found it more difficult than one would expect to find suitable citations for these seemingly obvious statements. Nevertheless, we agree that backing-up these claims with references would be beneficial and thank the referee for providing two citations. We added one of them, together with another reference, more closely related to experimental soil studies (Katagi, 2013). [Line 27]

*R1:* Line 23: List these three-dimensional characteristics – are these vegetation, hydrological, soil-mechanics, etc.

**Answer:** We listed some examples: „*They are shaped by physico-chemical processes and biological activity and, thus, have developed three-dimensional characteristics (e.g., vertical gradients, macropore network, root system, etc.) that cannot easily be reproduced artificially.*". [Lines 26-27]

*R1:* I feel as though Line 39, 'The collection of large monoliths (over 1m3) necessitates heavy machinery...' should be stressed in the Abstract, as this is a key reason for the study methods.

**Answer:** We rephrased parts of the Abstract to stress this more clearly: "*Acquiring large monoliths necessitates heavy machinery, which is time-, cost- and labour-intensive. However, small- to medium-sized soil blocks, however, can be obtained using less demanding methods.*". [Lines 9-11]

*R1:* Line 38: 'Advisable to use as large soil slabs as possible' – but this is quite generalised in the context of soil monoliths. How large, and when you refer to a soil 'slab', does this incorporate depth, width, and length? A reference here would be helpful to appreciate the scale or standard to use here.

**Answer:** The range of undisturbed monolith dimensions to be implemented in indoor studies is restricted to a few m width and depth, probably around 10 m² in area and depths < 2.5 m, even if very heavy gear is used. We do not think that it is feasible to provide a number or scale here above which a monolith is supposed to produce results that are representative.
However, we changed the wording here, so that it is clear that all three dimensions are meant and to avoid confusions about the term '*slab*' (which was used as a synonym for monolith/block). The wording now reads as follows: "*For upscaling purposes and a more accurate representation of soil processes and their variability, it would be desirable to use soil volumes that are as large as possible; however, the amount of work needed increases substantially with size.*". [Lines 40-42]

*R1:* Line 42: 'Blocks below 300 kg are easier to handle...' – a reference here is provided, but more context would be helpful on why this figure. Is this because it can be handled using a particular type of machinery. This could relate to the previous comment – maybe 300 kg is a good balance between being small enough that it is practical and easy to work with and large enough that edge effects are minimised. Some extra discussion on this point would be helpful.

**Answer**: The 300 kg were taken from the reference. We realise that this number appears to be quite arbitrary without context. We rephrased the sentence to a more generalised statement: "*Furthermore, the less heavy the monoliths are, the easier they are to handle, store, and discard*.". [Lines 47-48]
We do not think that the manuscript would be improved by further discussion. It is obvious that smaller and lighter monolith are easier to handle and store, but how easy the handling is also depends on the resources of the laboratory (access to cranes, lift trucks, manpower, etc.).

*R1: Line 44: 'Combination of two or more…' – are these equally sized blocks? Change to a 'combination of two or more equally sized, smaller monoliths'. State here that the material should also be comparable, as presumably, you wouldn't want to combine different materials.*

**Answer:** We were thinking that the monoliths would all be taken from the same site and with the same sampling gear and method, thus being equally sized. We rephrased the sentence to include this: "*A promising approach for runoff research appears to be taking two or more equal sized smaller monoliths at a site and combining them into a single large block, thereby optimizing cost and labour efficiency as well as representativity and upscaling potential*.". [Lines 49-51]

*R1: Line 49: Multiplication symbol rather than 'x'. Add in 'equally sized monoliths' before the dimensions.*

**Answer:** Changed. [Line 55]

*R1: Line 54: 'Done properly – the recombination procedure has no directional effect on runoff properties'. Discussing the correct procedure or recombination methods would be beneficial to support this statement.*

**Answer:** We agree and will added some more details on a '*proper re-combination*' in the M&M section: "*For a proper re-combination, the blocks must be of equal width and depth. Furthermore, the monoliths, and the soil used for the soil-water mixture, need to be taken from the same site close to one another*.". [Lines 76-78]

*R1: Line 55: Wording unclear '…do not differ regarding the (share of) outflow at…'. Please clarify the wording here. I think you are referring to separate outflows on combined monoliths, but the wording could be changed to make this clearer to the reader.*

**Answer:** What we wanted to express is that re-combined monoliths do not differ regarding the amount of outflow of the different flow pathways, as well as their proportional share. We changed the wording for clarification: "*Accordingly, we hypothesized that (1) re-combined monoliths do not differ regarding the amount of outflow at the different flow pathways or their proportional share, […]*". [Lines 61-62]

*R1: Line 57: Runoff velocity? Please clarify as this is in the supplementary material to the main text…are you expressing runoff as a speed m s-1, or is it discharge (volume/time). This needs to be made more apparent here.*

**Answer**: The unit of a velocity is distance per time (usually $m\ s^{-1}$), the unit of a discharge is volume per time. As we speak of '*runoff velocity*' we do not think that there is an ambiguity. However, we added the unit for clarity. [Line 112]

**Methods**

*R1:* Line 71: '…the soils occasionally dried up to some extent at the surface' – this is quite vague. I would consider removing this or putting more detail on this point.

**Answer:** We agree and have removed this sentence.

*R1:* Line 77: I don't know whether SRF is a good abbreviation for surface runoff. Also, please define what laterally exported water refers to. I don't think the word 'exported' is clear – maybe this should be reworded to lateral flow or lateral spread/transmission. You mention these abbreviations later in the text (Line 120), so I would either remove these abbreviations from the text/tables altogether or not refer to the longer wording in the text again, as these have already been defined in Line 77. I would recommend removing these abbreviations from the text/tables for clarity.

**Answer:** We agree. We removed all abbreviations for the flow pathways and instead use the full wording throughout the text and tables.
We also added a better description for laterally exported water, which now reads as follows: "*The runoff experiments were carried out in two experimental sets. During the first set, two flow pathways were recorded at the lower end of the monolith, the surface runoff and subsurface interflow. For the second set, we further sampled and distinguished between percolating water that went through the whole soil body vertically and water that infiltrated into the soil body, but left the monolith again at the side due to lateral flow pathways.*". [Lines 85-89]
Furthermore, we will change the wording from "laterally exported water" to "*lateral flow*" throughout the text as suggested. [Line 89 ff.]

*R1:* Line 85: Why were the slopes of 3% and 4% decided upon, and why did these change between runs.

**Answer**: Based on our data from established VFS in Austria, 3-4 % are commonly found slopes of VFS. We added some text to clarify this: "*The inclination was based on typical slopes of VFS in Austria and was adjusted to 3 % during the first set and 4 % during the second set.*". [Lines 96-97]
The first monolith of the second set was adjusted to 4 % because we erroneously thought we also used 4 % during the first set. The 4 % were than kept for the rest of the second set to have equal conditions within a set. As the switch from 3 to 4 % is rather negligible, we would refrain from further explanations in the text.

*R1:* Line 90: 'Fully saturated and left to dry for 24 hours' – why? To reach field capacity? Please justify does this relate to a standardised methodology (i.e. BS).

**Answer**: The monoliths were saturated (and left to drain) in order to reach comparable soil moisture levels for all monoliths. The aim of the drainage procedure was to obtain conditions of field capacity for the soil water content. Previous work has shown that for this type of soil monoliths (soil depths of about 30 cm), field capacity is reached after 1 day of free drainage (Tiefenbacher et al., 2021).
We added a clarification, which reads as follows: "*Before each experiment, the monoliths were transferred to a water pool until fully saturated and then left to drain for 24 h to obtain comparable field capacity conditions for the soil water content (see Tiefenbacher et al., 2021).*". [Lines 102-103]

*R1:* Line 92: 'A constant flow of 5 l/m was adjusted via a valve and water meter' – reword this to …' was applied using a valve and water meter'.
Line 92: Remove 'we used' and change it to 'deionised water spiked with ortho-phosphate was applied…'

**Answer:** We changed both sentences as suggested. [Lines 105-106]

*R1: Line 93: When you refer to 'mimic agricultural runoff', why these concentrations? Are they specific to regulations in Austria for farming practices?*

**Answer**: This concentration was based on a previous (unpublished) study, which analysed the runoff from fields using artificial rainfall. Data for typical (Lower) Austrian agricultural soils ranged from 0.2 – 0.9 mg DP $l^{-1}$. The used value of 0.5 mg $l^{-1}$ is more or less a mean value. We will add the wording "*typical local*" to clarify this a bit better. [Line 106]

*R1: Line 106: Reword 'we conducted' and subsequent use of 'we' pronoun.*

**Answer:** We changed the wording accordingly. [Lines 120-128]

*R1: Line 108: 'We used the quotient of outflow (of the respective flow pathway) to actual inflow' – please reword for clarity.*

**Answer:** Sentence was rephrased for clarity: "*To account for slightly different inflow rates between monoliths, a standardized outflow value was calculated, by dividing the measured outflow rate at a flow pathway by the actual inflow rate*.". [Lines 123-124]

*R1: Line 114: Are the scripts in the Appendices/supplementary materials?*

**Answer**: We did not introduce any new formulas or functions. We only used already existing libraries and the functions therein. Thus, we do not see much benefit for the reader of including the scripts to the Appendix. Nevertheless, as stated in the Section "*Code and data availability: [..] raw data and codes are available upon reasonable request from the corresponding author*." [Lines 323-324]

**Results**

*R1: Line 121: How much earlier? State this in the text.*

**Answer:** We added mean numbers to the text. The rephrased text reads as follows: "*Both treatments had a similar beginning of surface runoff outflow at around 69 s, but re-combined blocks had a 38 s earlier onset of lateral flow and 35 s later onset of percolating water on average.*". [Lines 136-138]

*R1: Line 134: High heterogeneity in the data – what might cause this? Some extra discussion on this would be helpful.*

**Answer:** We added a more detailed discussion on the sources of soil variability to the Discussion section 4.2, which reads as follows: "*Sources of this natural variability of the soil are manifold, including vegetation patterns, edaphon activity, (micro-)relief, soil aggregation, and their often complex interactions with runoff (Boix-Fayos et al., 2006; Bryan and Luk, 1981). Furthermore, there may be anthropogenic impacts before, during, and after the monolith collection (Luk and Morgan, 1981; Rüttimann et al., 1995).*". [Lines 247-250]

**Discussion**

*R1: Discussion is good. Because this is a technical note, do you have any 'lessons learned' or reflections that you could add if someone tried to replicate these experiments, how would they be able to improve their design?*

**Answer:** Some considerations can already be found in the Discussion section 4.1, to which we added additional paragraphs, (e.g., regarding what is necessary for a successful combination; different soil types, etc.): "*For a successful combination, the single blocks must be of equal width and height. While the sampling device usually fixes the width, acquiring similar heights may be more difficult. Uneven soil surfaces produce small barriers or ridges when monoliths are combined, which can interfere with runoff patterns. Furthermore, monoliths will have different heights at the front and back end if they are not taken at a right angle to the soil slope. Some adjustments can be made in the laboratory (e.g., different bottom plates to even height differences). Nevertheless, we strongly advise avoiding such issues in the first place by an a priori assessment of site conditions and careful sampling. This also includes a sampling design that allows the extraction of monoliths close to one another, limiting the effects of more significant scale gradients.*
*In principle, we expect the combination method to apply to a large variety of soil types, but we cannot yet provide textural limits. However, combining two soil blocks requires removing the bordering at one side. Thus, structurally weak soils that could collapse without a frame are not suitable for this method*". [Lines 188-197 ff.]

Note that we also deleted some sentences and paragraphs from the discussion following suggestions by other reviewers.

*R1:* Line 167: 'Blocks need to be watered regularly to avoid drying up' – please state why this is important. Is this because drying up will cause the blocks to dissociate from each other and thus affect the experiments, or is there another reason for this?

**Answer**: Drying would change the porosity within the soil, which affects runoff properties, but would also impede a proper 'repair' of the contact zone in combined monoliths. In addition, it will negatively affect plant growth of the monoliths.
We added a sentence to clarify this: "*Blocks need to be watered regularly for plant vitality and to avoid drying and an emergence of cracks in the soil structure that would affect runoff properties and impede the repair of the combined monoliths.*". [Lines 198-200]

*R1:* Line 191: 'Nevertheless, there was a trend of higher LAT outflow at re-combined blocks – does this refer to the monolith as a whole or just the interface at the contact area between the combined monoliths.

**Answer**: This refers to the whole (re-combined) block. We can only speculate what happens at the interface at the contact area (what we do in the following paragraph).

**Conclusion**

*R1:* The conclusion is reasonable, but given the number of tracer experiments that the monoliths were subjected to in the manuscript results/discussion sections, I would summarise these key findings in the conclusions. Specifically, what were the take-home messages of this study and what are the implications of each experiment? For example, recombining monoliths to conduct salt tracer experiments. Bare in mind that this is a technical note when summarising the conclusions.

**Answer**: We understand that the importance of our study is the finding that the method we used is a valid procedure. In this respect, the results of the different experiments are just a way to test the procedure and are not of primary interest as such. We have structured the take-home messages: in the first paragraph, we state what is important for the sampling and combination procedure; in the second paragraph, we describe why we think the results support the application of this method. Thus,

we still think that the take-home messages are outlined sufficiently and are fit for a Technical Note. If we are missing something, we kindly ask R1 for further clarifications.

**Figures/Tables**

*R1: Figure 1: Annotate arrows to show slope direction (from overflow tank to outflow at the bottom) for clarity. Also, it would be helpful to annotate key hydrological processes (surface runoff, sub-surface interflow, percolation, laterally exported water) as arrows.*

**Answer:** We changed Fig. 1 accordingly:

[Figure]

*Figure 1: Fig. 1: Setup of runoff experiments with flow pathways (arrows). [A] Overflow tank; [B] metal frame with surface runoff (RUN) collector; [C] metal plate for subsurface interflow (INT) collection; [D] bottom frame with collectors for percolating water (PER; inner outlets) and lateral flow (LAT; outer outlets); [E] rack with slope-adjustable gear.*

*R1:* Table 1: Reword caption 'Site and material characteristics. Add size factions for clay, silt and sand (i.e. > 2.0 mm) – are these according to BS or an international specification.

**Answer:** We reworded the caption and added size factions. The size factions are based on national standards (ÖNORM). The revised table header reads as follows: "*Table 1: Site and material characteristics. TOC – total organic carbon; CaCO$_3$ – calcium carbonite. Coarse material > 2 mm; Sand 2–0.063 mm; Silt 0.063–0.002 mm; Clay < 0.002 mm.*". [Lines 81-83]

*R1:* Table 2: Confusing layout…? Maybe plot the water budget…?

**Answer**: Plotting the water budget would result in an additional figure. We think that the advantage of providing this information in one table row is about saving space. Thus, we suggest keeping the table as it is.

---

## Author Comment (AC5)

**Response to Reviewer 2**

We would like to thank Reviewer 2 for the comments and remarks which significantly improved the manuscript. Please find our revisions and replies to the general and particular comments below. The line numbers in blue square brackets refer to the line numbers in the uploaded revised manuscript with tracked changes.

*R2: Soil characteristics (Table 1): TOC is undefined. Key soil characteristics such as porosity, dry density, gravel content, organic matter should be given if they are available. Could this method be applied to other soil types?*

**Answer:** We defined TOC and $CaCO_3$ and revised the table header, also following suggestions from another reviewer: "*Table 1: Site and material characteristics. TOC – total organic carbon; $CaCO_3$ – calcium carbonite. Coarse material > 2 mm; Sand 2–0.063 mm; Silt 0.063–0.002 mm; Clay < 0.002 mm.*". [Lines 81-83]

We added information on coarse material (> 2 mm). Unfortunately, we cannot provide data on the other requested soil characteristics.

In general, this method should be applicable to a rather broad spectrum of soil textures, which is the main criteria for application. However, we do not have information yet to add textural limits, but we suppose that soils with a very sandy texture will be difficult to handle, because of their weak structural integrity.

We added a discussion to section 4.1., which reads as follows: "*In principle, we expect the combination method to apply to a large variety of soil types, but we cannot yet provide textural limits. However, combining two soil blocks requires removing the bordering at one side. Thus, structurally weak soils that could collapse without a frame are not suitable for this method.*". [Lines 195-197]

*R2: Statistical analysis: differences between experiments are analyzed using statistical metrics. The Methods section does not clearly explain how this is done. For example, H and P (Line 132) are undefined. L. 114: Significance of what? What would have been the results at the 0.01 level?*

**Answer:** *H* and *P* are the outcomes of the Kruskal-Wallis test: *H* is the test statistic, *P* is the probability measure. These two abbreviations are commonly used and we suggest not to give definitions for these test statistic or p-value.

Statistical significance was set to 5 %, as is usually the case in virtually all studies across various scientific fields. A significance level, or $\alpha$, of 0.05 refers to a risk of 5 % that we conclude we have found significant differences between two (or more) groups when in fact there are none. This value is common practice, although, of course, arbitrary and sometimes criticized. We suggest not to deviate from it. Nevertheless, in the manuscript we do not solely rely on statistical significance alone, but also discuss the meaningfulness of found differences (e.g., if there is a directional effect or trend, how substantial the variation is, etc.). Our conclusions would have been no other if $\alpha$ would have been set at 0.001 (or any other value, for that matter).

We hope that we did not misunderstand the intention of R2 in this comment; if so, we would kindly ask R2 to clarify the issue.

*R2: Experimental design is unclear. For example, on L. 116, a "second experimental set" is mentioned. What is this? The first line of Table 2 is not complete for understanding the experimental plan. A new Table listing all experiments would be useful.*

**Answer:** The "*second experimental set*" is the one that is described in the manuscript; data on the first experimental set is provided in the Supplement. Both are described at the beginning of section 2.2. Maybe the wording ("*set*") is misleading? What we meant is synonymous to "*trial*" or "*round*". During the first set, we conducted experiments on all 6 monoliths and recorded the outflow at 2 flow pathways. After some improvement of the overall set-up, the second set started, during which we conducted another round of experiments on all 6 monoliths, but this time recording the outflow at 4 flow pathways. As the latter is more elaborate, we decided to only report on the results of the second set in the main text. However, as stated in the discussion, the first set provides similar results. Thus, as long as not explicitly stated, all results in the main text refer to the second experimental set.

To clarify this, we rephrased the paragraph, which reads as follows: "*The runoff experiments were carried out in two experimental sets, each comprising a full round of experiments (runoff and tracer measurements) on all six monoliths. During the first set, two flow pathways were recorded at the lower end of the monolith, the surface runoff and subsurface interflow. For the second set, we further sampled and distinguished between percolating water that went through the whole soil body vertically and water that infiltrated into the soil body, but left the monolith again at the side due to lateral flow pathways.*". [Lines 85-89]

Maybe we are missing the point; in this case we would kindly ask R2 for a clarification.

*R2: Discussion: Section 4.2 is too long, especially for a technical note. Could be more concise.*

**Answer**: We agree that the discussion is long, especially for a Technical Note. This is mainly due to the fact the we used a rather extensive approach (outflow measurements, chemical loading of runoff, flow velocity). We shortened and rephrased the discussion where possible to be more concise. Parts of the Discussion were also changed/rephrased following suggestions from other reviewers.

*R2: The title of the paper could be improved.*

**Answer**: We agree, and changed the title to "*Technical Note: Combining undisturbed soil monoliths for hydrological indoor experiments*". [Lines 1-2]

---

## Author Comment (AC6)

**Response to Reviewer 3**

We would like to thank Reviewer 3 for the comments and remarks which improved the manuscript. Please find our revisions and replies to the general and specific comments below. The line numbers in blue square brackets refer to the line numbers in the uploaded revised manuscript with tracked changes.

**General comment / main concern:**

*R3: My only main concern is about the experimental design. While the overall research question is about constructive large monoliths e.g., over 1 m³ (L39) that should obtain more accurate representation of soil processes, the experiments are conducted to smaller size blocks (1 x 0.5 x 0.35 m). In addition, only two tests are conducted and with same initial and boundary conditions (constant flow of 5 l min-1 after 24h from fully saturated soil). Finally, as different monoliths are compared, soil heterogeneity affect the interpretation of the results. So, how could we address the questions and reach any conclusions? In contrast, based on the research question and the overall general introduction of the study, I would have expected to see, e.g.: tests conducted on larger combined monoliths (e.g., two blocks of 1 x 0.5 x 0.35 m); tests conducted at the same monolith uncut and cut; tests conducted with different initial and boundary conditions. Field tests over the same area could have also been be performed for comparison. For this reason, I would strongly suggest to increase the number of experiments to try to derive some more general conclusions. In case these could be not anymore feasable, I suggest to rephrase several parts of the manuscript and weakening some statements as listed in the specific comments below.*

**Answer:** In general, we agree with what the reviewer states in this comment. Conducting experiments on larger monoliths, using different initial and boundary conditions, or the inclusion of field tests would have all been beneficial to support our hypothesis. However, we see no solution to the problem of soil heterogeneity other than repeating experiments with a high number of replicates. In this way, we may at least find out if soil heterogeneity is larger than procedural heterogeneity (which we did). Please be aware, that using the same soil monoliths uncut and cut will also introduce heterogeneity of unknown size due to the handling necessary during the resampling procedure.

Indeed, as R3 already mentions, conducting these additional experiments was/is not feasible. In fact, our experimental set-up presented in this manuscript (6 monoliths, 2 experimental sets, analysis of runoff characteristics, chemical properties, and flow velocity, 4 flow pathways) was already quite time-consuming and labour-intensive. While we agree that the suggested additional experiments are reasonable additions and something that may be done in future studies, we think that our results are sufficiently informative to draw the conclusion that combining monoliths is a sound procedure. However, we discuss this issue in more detail in the manuscript and rephrased several parts of the manuscript as suggested by R3 and the other reviewers (see Responses to Specific Comments).

**Specific comments**

*R3: Title not really meaningful, please consider rephrase. The two worlds should be field and lab but no experiments are actually conducted to answere that a combined monolith is the best of both. See general comment.*

**Answer:** We agree and rephrased the title to "*Technical Note: Combining undisturbed soil monoliths for hydrological indoor experiments*". [Lines 1-2]

*R3: L7 I guess there are several major decision in soil hydrology depending on the objectives and expertise. The statement that a major decision in soil hydrological research is whether to conduct experiments outdoor or indoors is in my opinion a bold statement. Please consider rephrasing.*

**Answer**: We agree that the decision whether to conduct the experiment under field or laboratory conditions is one of several options in soil hydrology. We rephrased the sentence: "*An important decision […]*", to better reflect this. [Line 7]

See also response to Comment L19.

*R3: L16. The study can be improved by adding some suggestions about which experiments should be conducted for a definite conclusions, e.g, block size, initial and boundary condition, comparison to field tests etc.*

**Answer**: We deleted this sentence from the Abstract (following other reviewers' suggestions). We think that the wording (*our results suggest…, we propose*…) is defensive enough to clarify that there is still room for improvement. [Lines 17-19]
Nevertheless, we discuss this issue in more detail in section 4.2.: "*We used a comprehensive approach, analysing runoff characteristics, chemical loadings, and flow velocity and are, thus, confident in our conclusions. Nevertheless, there is room for further research, for instance applying different boundary and initial conditions, such as other soil types and flow rates, or a higher sample size and number of blocks to be combined. A comparison of large monoliths taken with heavy machinery with blocks of similar extent that are made up of combined smaller monoliths would be very interesting.*". [Lines 297-301]

*R3: L19. Is this really a cardinal question? For some reasons one could always be fine with lab or field test.*

**Answer**: We agree that one could always be fine with lab or field tests. However, each comes with specific advantages and drawbacks and they are, thus, not interchangeable and dependent on the research question and resources. The rather strong (original) wording of this sentence has also been driven by the fact that in soil hydrology (as opposed to other hydrological fields) the question of conducting indoor or outdoor studies is quite often an issue.
To account for this comment, we changed the wording from "*cardinal*" to "*important*". [Line 22]

*R3: L68 The blocks were interchanged so that the left front of the left block faced the right front of the right block. Why?*

**Answer**: Blocks were interchanged to better mimic the combination of two separately taken monoliths. If we would have combined the separated monoliths again directly at the cut without interchanging we were concerned that this would lead to a "perfect fit" (e.g., concerning macropore channels) that would not be possible to obtain when separately taken monoliths are used.

*R3: L71. Discussion can be extended considering the results we should expect for different soil type*

**Answer**: We added a discussion on different soil types at section 4.1.: "*In principle, we expect the combination method to apply to a large variety of soil types, but we cannot yet provide textural limits. However, combining two soil blocks requires removing the bordering at one side. Thus, structurally weak soils that could collapse without a frame are not suitable for this method.*". [Lines 195-197]

*R3:* L92. Why this flow? Is this representative of hydrological conditions? Alternative scenarios should have been performed. If not possible, discussion should be extended with suggestions about.

**Answer**: This flow rate was chosen based on previous experiments. We know from previous (unpublished) studies on similar soils that a large (though not extreme) rainfall event of 60 mm would produce a runoff within the range of 0.1 l min$^{-1}$ m$^{-2}$. The chosen 5 l min$^{-1}$ would therefore correspond to a source area of 500 m² that contributes to a concentrated flow at the field edge – which is probably at the lower end of realistic source areas. Other studies using artificial runoff (e.g., Guertault et al. 2021, Saleh et al. 2017) used much higher flow rates (12-99, and 99 l min$^{-1}$, respectively). This would not be possible to handle with our set-up. Nevertheless, we also chose a lower flow rate to encourage infiltration, as we speculated that the cut could act as a large macropore. Higher flow rates and, thus, higher flow velocities would promote surface runoff at the expense of infiltration.
Suggestions on experiments with different flow rates are included in the added paragraph referred to in the Response to Comment L16 (see above).

*R3:* L205. The fact that spatial heterogeneity affects more than cut monolith can not be considered as a proof of the validity of the combined procedure.

**Answer**: We agree that the high heterogeneity encountered is not a proof of the validity of the combination procedure. It is still possible that there is an effect of the combination, which would probably require a (maybe unfeasibly) high number of plots (replicates) to find out. Based on our data, even if there is an effect it would be small in comparison to the soil heterogeneity. Based on our findings, we only state that "… *there is no indication to reject the hypotheses*", which is true. Thus, we think that our conclusion that the advantages outweigh potential disadvantages of added data noise is supported.

*R3:* L213-214. Combining more monolith has been not proofed to be valid in the present study

**Answer**: We agree and rephrased and weakened this statement as follows: "*We conclude that the re-combination procedure did not lead to directional differences and, thus, had no adverse effect on runoff properties, suggesting that combining two (and probably more) blocks is a viable and practicable way to obtain single larger soil monoliths.*". [Lines 253-255]

*R3:* L230. What is Video 1?

**Answer**: Video 1 is part of the *Assets*. The *Assets* consist of a *Supplement* and a *Video supplement* and should both be downloadable at the Preprint Website (Copernicus Office). This short video shows a concentrated outflow at a single outlet, probably an earthworm channel.

*R3:* L244. During drying the cut seems to become relevant. So, it could be argue that also runoff could be affected depending on boundary conditions.

**Answer**: We agree; consequently, it is important that the monoliths are properly maintained, especially regarding regular watering. Experiments should not be carried out on dry-ish soil conditions. We added clarifications such as: "*We recommend that soil monoliths are kept outside in a sheltered but sunny location. Blocks need to be watered regularly for plant vitality and to avoid drying and an emergence of cracks in the soil structure that would affect runoff properties and impede the repair of the combined monoliths*", and

"*Furthermore, proper storage and maintenance, especially regular watering, are crucial to keeping the monoliths in good condition and are, in turn, dependent on the research aim, the duration of the*

*experiment, climate, and resources (e.g., staff, storage space).*", to the Discussion and Conclusion. [Lines 198-200; 306-308]

*R3: 261. Same comment as for L213-214*

**Answer**: We think R3 is referring to the sentence "… *we recommend the use of combined monoliths* …". We admit that the term "*combined monoliths*" may be a bit ambiguous if two or more monoliths are meant. We suggest to leave this unchanged, but we added some more information in the following sentence to clarify this issue: "*Nevertheless, we encourage further research on this subject to better delimit the potential and possible limitations of this procedure, for instance using different experimental setups (e.g., number of monoliths), boundary conditions (e.g., flow rates, soil types, dimensions), or analysis methods (e.g., X-ray imaging).*". [Lines 317-320]